# Target Enrichment Metagenomics Reveals Human Pegivirus-1 in Pediatric Hematopoietic Stem Cell Transplantation Recipients

**DOI:** 10.3390/v14040796

**Published:** 2022-04-12

**Authors:** Natali Ludowyke, Worakorn Phumiphanjarphak, Nopporn Apiwattanakul, Suwimon Manopwisedjaroen, Samart Pakakasama, Insee Sensorn, Ekawat Pasomsub, Wasun Chantratita, Suradej Hongeng, Pakorn Aiewsakun, Arunee Thitithanyanont

**Affiliations:** 1Department of Microbiology, Faculty of Science, Mahidol University, Bangkok 10400, Thailand; nataliroseludowyke@student.mahidol.ac.th (N.L.); ac@wphumi.com (W.P.); swiboonut@gmail.com (S.M.); 2Pornchai Matangkasombut Center for Microbial Genomics, Department of Microbiology, Faculty of Science, Mahidol University, Bangkok 10400, Thailand; 3Department of Pediatrics, Division of Hematology and Oncology, Faculty of Medicine, Ramathibodi Hospital, Mahidol University, Bangkok 10400, Thailand; nopporn.api@mahidol.ac.th (N.A.); samart.pak@mahidol.ac.th (S.P.); suradej.hon@mahidol.ac.th (S.H.); 4Center for Medical Genomics, Faculty of Medicine, Ramathibodi Hospital, Mahidol University, Bangkok 10400, Thailand; insee.sen@mahidol.edu (I.S.); wasun.cha@mahidol.ac.th (W.C.); 5Virology and Molecular Microbiology Unit, Department of Pathology, Faculty of Medicine, Ramathibodi Hospital, Mahidol University, Bangkok 10400, Thailand; ekawat.pas@mahidol.ac.th

**Keywords:** human pegivirus, HPgV-1, hematopoietic stem cell transplantation, next-generation sequencing, viral enrichment

## Abstract

Human pegivirus-1 (HPgV-1) is a lymphotropic human virus, typically considered nonpathogenic, but its infection can sometimes cause persistent viremia both in immunocompetent and immunosuppressed individuals. In a viral discovery research program in hematopoietic stem cell transplant (HSCT) pediatric patients, HPgV-1 was detected in 3 out of 14 patients (21.4%) using a target enrichment next-generation sequencing method, and the presence of the viruses was confirmed by agent-specific qRT-PCR assays. For the first time in this patient cohort, complete genomes of HPgV-1 were acquired and characterized. Phylogenetic analyses indicated that two patients had HPgV-1 genotype 2 and one had HPgV-1 genotype 3. Intra-host genomic variations were described and discussed. Our results highlight the necessity to screen HSCT patients and blood and stem cell donors to reduce the potential risk of HPgV-1 transmission.

## 1. Introduction 

Two strains of pegiviruses are known to infect humans, namely the human pegivirus-1 (HPgV-1) and the human pegivirus-2 (HPgV-2; also referred to as Human Hepegivirus 1) [1]. Out of the two, HPgV-1 is more prevalent and was initially named GB virus type C or hepatitis G virus. It is a persistent lymphotropic positive-sense RNA virus, within the genus *Pegivirus* of the family *Flaviviridae* [2,3,4], and it is known to transmit by blood transfusions, parenterally, sexually, and from infected mothers to their newborn infants [3]. It can induce persistent viremia in humans, but its biology and clinical significance is poorly known [5]. It is often reported to be associated with liver and kidney transplantations and blood transfusions [4,6,7,8,9] and has even been shown to be present in the brain tissue of a patient with encephalitis [10]. 

In Thailand, several studies conducted before the year 2000 reported the prevalence of HPgV-1 among kidney transplant patients, thalassaemic children, patients with chronic liver disease, sex workers, and intravenous drug users [11,12], and more recently, in 2009, the transmission of HPgV-1 from mother to child [13]. However, over the past two decades, the prevalence of HPgV-1 in hematopoietic stem cell transplant (HSCT) pediatric patients has not been reported globally.

In this study, we used a target enrichment next-generation sequencing (TE-NGS) approach to identify viruses that might be present in the blood of HSCT pediatric patients. Our study detected HPgV-1 in 3 out of 14 patients, all of which were detected in patients who developed post-HSCT febrile neutropenia (FN; 13 patients), which was not covered by the routine PCR-based tests used in the present clinical setting in Thailand. The virus complete genomes were assembled and characterized, and intra-host genomic variations were described and discussed.

## 2. Methods

### 2.1. Patients and Samples

This study was carried out in one of the centers for bone marrow transplantation that belongs to the Department of Pediatrics, Faculty of Medicine, Ramathibodi Hospital, Mahidol University, Thailand. The patient cohort included 14 pediatric patients, aged between 1 and 12; 13 of them developed febrile neutropenia (FN) post-HSCT. All patients underwent HSCT between November 2018 and December 2019, and all were on immunosuppressive therapy before and after transplantation (Appendix A). After HSCT, they were observed for short-term clinical complications such as engraftment syndrome, veno-occlusive disease, prolonged fever, and graft versus host disease (GVHD), and long-term clinical complications (>2 years) such as disease relapse (Table 1).

Peripheral whole blood samples (3–5 mL) were collected from each patient on the day of admission (DA) for the transplantation procedure. Two additional blood samples were collected for those who developed FN, namely day 0 (D0) and day 3 (D3) samples. The D0 sample was collected on the day of FN onset, and the D3 sample was collected 3 days after the D0 sample collection, with the aim of verifying the presence of viruses in the D0 sample, if any. Both D0 and D3 samples were subjected to the routine multiplexed probe-based quantitative real-time polymerase chain reaction (qRT-PCR) for the detection of common blood-borne virus pathogens, and the TE-NGS analysis 24–48 h after the collection. The workflow is illustrated in Figure 1. DA samples were sequenced only if D0 or D3 samples were detected positive for viruses.

### 2.2. Nucleic Acid Extraction 

Cultured Newcastle disease virus (NDV), an RNA virus of non-human origin, was added to the samples at concentrations ranging between 1 × 10^3^ and 1 × 10^4^ particles per 1 mL of whole blood as a positive internal control (Appendix A). NDV preparation protocol can be found in Appendix A. Plasma and cells were separated using centrifugation, and red blood cells (RBC) were removed using RBC lysis buffer (Geneaid, Taiwan). Total nucleic acid extraction was extracted from 1 mL of plasma and white blood cells (WBC) using QIAamp ccfDNA/RNA kit (Qiagen, Hilden, Germany) and AllPrep DNA/RNA Mini kit (Qiagen, Germany) according to the manufacturer’s recommendations, respectively. Extracted RNA from both kits was reverse transcribed using SuperScript III kit (Invitrogen Life Technologies, USA). cDNA and extracted DNA was pooled prior to second-strand DNA synthesis using Klenow fragment (3′–5′exonuclease negative) (New England Biolabs, Ipswich, MA, USA). DNA samples were tested to confirm the presence of the internal positive control NDV, using HotStarTaq^®^ PCR kit (Qiagen, Germany) with SYBR-Green [14] (Appendix A). 

### 2.3. Library Preparation

DNA sequencing libraries were prepared following the SeqCap EZ HyperCap workflow user’s guide (Version 2.3, Roche, Switzerland). Up to 12 samples were pooled and hybridized to the VirCapSeq-VERT probe pool for 16 h (SeqCap EZ VirCapSeq-VERT Panel). The system consisted of ~2 million oligonucleotide biotinylated probes, designed specifically to capture and enrich genomic coding sequences of more than 200 vertebrate viruses [15].

### 2.4. Sequencing

The DNA library was denatured and diluted using the MiSeq reagent kit v3 (MS-102-3003, Illumina, San Diego, CA, USA) following the MiSeq system, denature and dilute libraries guide (Illumina, USA). All specimens were (paired-end) sequenced with single indexing on MiSeq (Illumina, USA).

### 2.5. Virus Identification

Sequencing data were demultiplexed and Q30 filtered by Illumina software to generate FastQ files for each sample. For virus identification, Virus Identification Pipeline (VIP) v.0.1.0 [16] was used with the default settings under the ‘sense’ mode. A virus detection was considered positive if its read count relative to the total read number was >0.01% and the coverage was >65% with respect to the reference genome in the VIP database. 

HPgV-1’s were identified in 3 patients (see Results) and the detections were validated using a previously described PCR protocol [17] with some modifications (Appendix A). The negative results, i.e., the absence of HPgV-1, were also confirmed using the same protocol. Total nucleic acids from the plasma of the DA, D0, and D3 samples (200 µL each) were extracted using GENTi Viral DNA/RNA Extraction Kit (GeneAll, South Korea), and SYBR-Green-based qRT-PCR was performed using One-Step RT-PCR Kit (Qiagen, Germany). Little is currently known about the prevalence of HPgV-1 in Thai children (see Discussion). Therefore, HPgV-1 was also screened for in 112 whole blood samples collected from healthy adolescents (HA) using the above-mentioned PCR protocol. These samples were leftover blood from a health check-up collected from students aged 12–18 years (grade 7–12) in a school, Ayutthaya province, Thailand.

### 2.6. Genome Assembly

Metagenomic short reads were first assembled into contigs by using metaSPAdes v.3.15.1 [18] and IDBA-UD v.1.1.3 [19] with 11 k-mer values, starting from 21 and increasing to 121 with an increment of 10 mers each time. The assembled contigs were further assembled into scaffolds by using Quickmerge v.0.3 [20] with the default settings. Scaffolds were orientated and ordered by comparing them to viral genome sequences available in the NCBI RefSeq database using BLASTn [21] with the maximum e-value of 1e-10, the blast word size of 32, and the maximum target sequences of 10. Short reads were then mapped back to the assembly using bwa-mem2 v.2.1 [22] with the default settings for the purpose of quality checking, variant calling, and consensus sequence reconstruction. LoFreq [23] was used to call variants with the following options: -q 20 -Q 20 -m 60 -C 5 -D -d 1000000 -a 0.1, and the consensus function in the bcftools v.1.10.2 program package [24] was used to reconstruct consensus sequences, using the preliminary assembled and ordered scaffolds as references. Polymorphic sites with aggregated minor allele frequencies of more than 20% and with at least 20× read-mapping depths were characterized. 

### 2.7. Phylogenetic Reconstruction

Phylogenetic analysis was performed to genotype the detected viruses. A multiple sequence alignment of the viruses detected and reference sequences obtained from the NCBI nucleotide database was made by using MAFFT v.7.453 [25]. Potential recombination regions within the alignment were checked by using RDP, GENECONV, Chimera, MaxChi, BootScan, SiScan, and 3Seq with their default settings, all implemented in RDP4 v.4.100 [26]. Regions detected by more than 4 programs were considered significant and were removed from all of the sequences in the alignment manually to make a recombination-free alignment. A maximum likelihood phylogenetic tree of the virus was reconstructed by using IQ-TREE 1.6.12 [27] with the best-fit nucleotide substitution model (GTR+F+I+γ4) as determined by ModelFinder [28] under the Bayesian information criterion. SH-aLRT and ultrafast bootstrap branch support was computed based on 10,000 bootstrapped trees. 

## 3. Results

### 3.1. Enrolled Patients

In total, 14 HSCT pediatric patients were enrolled for this study, and 13 of them developed febrile neutropenia (FN) post-HSCT. Whole blood samples were collected from each of these patients on the day of admission to the hospital (DA). For those that developed post-HSCT FN (i.e., all the patients except Patient 014), two additional whole blood samples were collected, one on the first day of the fever onset (D0) and 3 days after (D3). In one case (Patient 005), only DA and D3 samples were available. Demographic data of the patients, including age, gender, underlying disease, type of HSCT, and the given conditioning regiments can be found in Table 1. The majority of patients had low WBC counts, both on D0 and D3 (Appendix A). 

### 3.2. Virus Detection by Routine Tests 

Out of the 14 patients, 2 patients (Patient 004 and 008) were tested positive for cytomegalovirus (CMV) in both of their D0 and D3 samples by routinely performed multiplex probe-based qRT-PCR assay, which screened for human adenovirus, human cytomegalovirus, Epstein–Barr virus, herpes simplex virus 1, herpes simplex virus 2, varicella-zoster virus, enterovirus, human parechovirus, human herpesvirus 6, human herpesvirus 7, and human parvovirus B19. In both patients, threshold cycle (Ct) values were between 31.1 and 36.8 (Table 2).

### 3.3. Virus Detection by TE-NGS Technology 

To detect viruses in the blood samples that were not covered by the routine qRT-PCR assay, TE-NGS was applied, and VIP [16] was used to detect viral sequences within the NGS data produced. qRT-PCR was used to validate the bioinformatics results. The positive internal control, NDV, spiked at either the concentration of 1 × 10^3^ (Ct = 30.15–32.58) or 1 × 10^4^ (Ct = 25.64–27.47) particles per 1 mL of whole blood, was detected in all samples (Appendix A), demonstrating that this approach was sensitive. Averaged across all samples, 45.1% (1.7–88.7%), 35.6% (2.5–91.8%), and 19.3% (6.5–60.2%) reads were identified as human, virus, and unidentifiable sequences, respectively, by VIP (Appendix A). 

CMV was detected in Patient 004’s D3 sample (Ct of qRT-PCR = 31.18) by TE-NGS, but not in Patient 004’s D0 sample or Patient 008’s samples, in which the virus concentrations were much lower (Ct of qRT-PCR = 34.07–36.89).

In addition to CMV, HPgV-1 sequences were detected as positive in three patients’ D0 and D3 samples, including Patient 002, 011, and 015. The analysis also detected the virus in Patient 002 and 011’s DA samples (94.2% and 100% genome coverage, respectively, Table 2), consistent with that the patients likely acquired the virus before the hospitalization. The detections of HPgV-1 by bioinformatic analyses were confirmed by subjecting plasma samples to qRT-PCR using HPgV-1 specific primers [29], yielding Ct values ranging between 25.17 and 30.65 (Table 2). The fact that HPgV-1 could be detected in the plasma is suggestive of an active infection. On the other hand, HPgV-1 could not be detected in Patient 015’s DA sample by both NGS data analysis and qRT-PCR analyses, raising a possibility that the patient could have acquired the virus during hospitalization or the course of HSCT. None of the samples from Patient 014, who did not develop FN, and from 112 healthy adolescents were detected positive for HPgV-1 by qRT-PCR.

Among the three HPgV-1 positive patients, Patient 002 developed engraftment syndrome and had prolonged fever from day 1–16 post-HSCT, while none of them developed veno-occlusive disease or GVHD (Table 1). When it comes to long-term clinical effects, Patient 002 and 011 recovered normally, while Patient 015 experienced neuroblastoma relapse 5 months post-HSCT. 

### 3.4. HPgV-1 Genome Assembly

Whole genomes of HPgV-1 detected in the eight samples (three from Patient 002, three from Patient 011, and two from Patient 015) were de novo assembled; seven of which had a full-length coding sequence, while one lacked a short sequence on the 5’ end of the coding region (Figure 2). The assembly sizes ranged between 8714 bases and 9336 bases, with average depths of at least 154× (Figure 2 and Table 3).

Polymorphic sites with aggregated minor allele frequencies of more than 20% and with at least 20× read-mapping depths were determined by mapping NGS reads back to the consensus sequences. The average number of polymorphic sites per kilobase for Patient 002, 011, and 015’s HPgV-1s was 9.27, 3.86, and 0.80, respectively (Figure 2, Appendix A). Of all the polymorphic sites identified, 94.86% were either C/T variant sites (56.76–67.44%) or A/G variant sites (27.91–40%). 

### 3.5. Genotyping

The genotypes of the detected HPgV-1s were determined by using phylogenetic analysis. Based on their phylogenetic placements, the results suggested that those obtained from Patient 002 and 015 were genotype 2, while those in Patient 011 were genotype 3 (Figure 3). Although the viruses from Patient 002 and 015 were found to be sister taxa, the branches separating the two were long and the two patients were admitted to the hospital at different times (1 year apart), thus they were likely distinct viruses, and it was unlikely that they were a result of direct virus transmission between the two patients.

## 4. Discussion

Viral infections are common complications following HSCT, and 42.9–68.2% of the total documented post-HSCT infections are viral infections [30,31,32,33,34]. Viruses most commonly found in HSCT patients include Epstein–Barr virus, cytomegalovirus, human herpesvirus 6, human adenoviruses, and polyomavirus BK [33,34,35,36,37,38]. A multitude of clinical tests is used for virus detection. Among them, polymerase chain reaction (PCR)-based techniques are typically considered the gold standard due to their ability to rapidly and precisely amplify small amounts of viral sequences that might be present in a clinical sample [39]. In the current clinical settings in Thailand, PCR-based approaches are routinely used to detect viruses in HSCT patients, and in our case, detected CMV in two patients, namely Patient 004 and 008. 

One of the main limitations of PCR-based methods is that they require target-specific primers for pathogen detection, making their ability to detect novel or uncommon pathogens very limited. Because of this, a significant portion of viral infections might be overlooked, and thus, a sensitive diagnostic method that can detect a broader range of viruses, or ideally, any viruses that might be present in clinical samples, is a timely need. Viral metagenomics is a hypothesis-free approach that has shown promise for comprehensive characterization of human virome with no prior clinical knowledge required [40,41]. Due to typically low fractions of viral genetic materials in clinical samples, direct application of the NGS technology to clinical diagnosis has been challenging; however, studies have shown that an addition of a viral genetic material enrichment step to the protocol could offer a solution to this problem [42,43,44,45]. The VirCapSeq-VERT is a probe panel comprised of ~2 million probes that cover all the viruses known to infect vertebrates. Its potential to enrich viral sequences of known viruses up to 10,000-fold when compared to conventional high throughput sequencing, as well as to detect novel viruses with an overall nucleotide divergence in the range of 40% has been demonstrated [15]. This makes this probe panel suitable for detecting divergent and potentially unknown viruses in clinical samples, which is essential when it comes to analyzing clinical samples from patients with unexplained illnesses, and for outbreak investigation of both known and unknown infectious etiologies. Subsequent other studies have also utilized this panel to explore the virome of a range of clinical and environmental samples and have obtained compelling results [46,47,48,49,50]. 

Therefore, in the present study, we utilized an unbiased VirCapSeq-VERT incorporated TE-NGS approach to screen for viruses that might be present in the whole blood samples of HSCT patients. The NGS-based test detected CMV only in Patient 004, specifically in their D3 sample, which had a moderate amount of target sequences (Ct = 31.18), but not in Patient 004’s D0 sample or Patient 008’s samples, which had relatively much lower viral concentrations (Ct = 34.07–36.89). When detecting viruses with a low concentration, it has been discussed that the sensitivity of agent-specific qRT-PCR could be higher than that of a TE-NGS test [51]. In any case, since the Ct values were very high, alternatively, it was thus possible that the qRT-PCR results could be false positive. Further investigations are needed to examine the relative sensitivity and specificity of the two assays when viral concentrations are very low.

In addition to CMVs, HPgV-1 was detected in three patients by TE-NGS, and was validated by qRT-PCR, including Patient 002, 011, and 015. The virus was not covered by the routine qRT-PCR tests in our clinical setting. Our results suggested that Patient 002 and 011 likely acquired the virus before the hospitalization, while Patient 015 may have acquired it during the hospitalization, but without any noticeable clinical symptoms that might be indicative of the virus acquisition. Due to the patients’ multiple blood transfusion histories, and the lack of data from their stem cells donors, the precise source of infection of the three patients cannot be concluded.

All of the three HPgV-1 positive patients developed FN after the HSCT procedure. The average time taken to develop FN in HPgV-1 positive patients was slightly lower than that of HPgV-1 negative patients, estimated to be 3.3 days and 4.7 days, respectively. We, however, feel that the number of the patients in our study was probably too small to make any conclusive statements regarding (apparently positive) association between HPgV-1 and the early onset of FN. Regarding other post-HSCT complications, Patient 002 developed engraftment syndrome and prolonged fever up to 16 days post-procedure, while none of the three developed veno-occlusive disease or GVHD. It is nonetheless worth noting that such post-HSCT complications could be observed in several HPgV-1 negative patients as well (Table 1). As for the long-term complications, we observed one of the three patients detected positive for HPgV-1 (33%), namely Patient 15, to experience a disease relapse 5 months after the transplantation. In the HPgV-1 negative patient group, 3 out of 11 patients (27%) were found to experience long-term clinical complications (Patient 008: chronic GVHD; Patient 013: died from GVHD and bleeding, and Patient 014: died from respiratory distress syndrome 39 days post-operation; Table 1). Despite this observation, again, we feel the sample size herein was probably too small to establish a link between HPgV-1 and long-term clinical complications.

We estimated the prevalence of HPgV-1 in this patient cohort to be 3/14 = 21.4%, comparable to those reported by other studies done in adult HSCT patients, ranging between 18% and 42% [52,53,54,55]. However, none of the studies reported the whole genomes of the identified viruses. Despite the reported high prevalence of HPgV-1, it is not listed among the viruses currently regarded as most relevant in the HSCT setting [33,36,37]. In contrast, we could not detect HPgV-1 in any of the 112 healthy adolescents samples. This suggested that the prevalence of HPgV-1 in our HSCT pediatric cohort could be much higher than that of the young healthy Thai population. However, since all 112 samples were from students of a single age group from a single school, the results should thus be interpreted with care and should not be extrapolated to general Thai children. To the best of our knowledge, there have not been reports on the prevalence of HPgV-1 in the Thai general population in the past 15 years. However, the prevalence of HPgV-1 in the Asian general population has been reported to be typically <10% [56], indeed, lower than those reported for HSCT patients [52,53,54,55], consistent with our speculation. 

In general, the risk of HPgV-1 infection has shown to be high in those who are exposed to blood and blood products, those on hemodialysis, and patients with chronic hepatitis C or human immunodeficiency virus-1 (HIV-1) infection [1]. Interestingly, multiple studies have reported that persistent HPgV-1 infection may have an immunomodulatory effect in patients co-infected with HIV-1, slowing the disease progression of acquired immunodeficiency syndrome (AIDS) and prolonging the survival [reviewed in [57]]. Therefore, it is not only important to screen the general Thai population, but also patients with other diseases, particularly HIV-1 patients, who are relatively prevalent in Thailand [58]. 

Up to now, seven HPgV-1 genotypes have been described [59,60]. Our phylogenetic analyses suggested that the genotypes of the HPgV-1s detected in the three patients were different; the ones infecting Patient 002 and 015 likely belonged to genotype 2, while that in Patient 011 likely belonged to genotype 3 (Figure 3). Two previous studies have also reported the detection of genotype 2 and 3 in Thailand [13,61], in addition to that, one of those studies has reported the detection of genotype 4 [13]. On a global scale, genotype 2 is most prevalent in Europe and North America but also can be seen worldwide, while genotype 3 is endemic in Asian countries and Amerindian populations [9]. 

Within the host, viruses typically experience very strong immune selection pressure, which drives them to diversify resulting in a cloud of genetic variants [62]. Our analyses showed that Patient 002’s HPgV-1 had the highest number of polymorphic sites, while Patient 015’s HPgV-1 had the lowest number of polymorphic sites (9.27, 3.86, and 0.80 polymorphic sites per kilobase on average for Patient 002, 011, and 015, respectively) (Figure 2, Appendix A). The low number of polymorphic sites in Patient 015’s viruses are suggestive of a relatively recent acquisition of the virus, consistent with the fact that Patient 015’s DA sample was HPgV-1-negative. 

Of all the polymorphic sites identified, 94.86% were either C/T or A/G variant sites, consistent with the fact that transition mutations are generally more likely to occur than transversions chemically. Besides polymerase and sequencing errors, host RNA editing mediated by ADARs (adenosine deaminases acting on RNA) may play some roles in the observed high numbers of transition changes as well. ADARs catalyze deamination of As to Is on double-stranded RNA substrates, which subsequently results in base changes from As to Gs or Us to Cs on the single-stranded RNA virus genome [63,64]. Many studies have shown that ADARs can induce hyper A-to-G and U-to-C mutations in a wide range of RNA viruses [64,65,66,67,68,69]. 

Interestingly, there were substantially more C/T variant sites (56.76–67.44% of the total number of all polymorphic sites characterized) than A/G variant sites (27.91–40%), and the ratios of C/T variant sites to A/G variant sites were similar across all samples (average ratios of normalized C/T variant sites to A/G variant sites: Patient 002: 1:0.57; Patient 011: 1:0.51; and Patient 015: 1:0.56) (Appendix A). This observation could potentially be explained by C-to-U hypermutation mediated by members of the apolipoprotein B mRNA-editing enzyme, catalytic polypeptide-like (APOBEC) family, which, unlike ADARs, catalyze deamination of Cs to Us directly on single-stranded nucleic acid molecules. Antiviral activities of APOBEC enzymes have been well demonstrated in retroviruses, retroelements, and some DNA viruses [70], but recent studies have suggested that it may restrict RNA viruses as well, in particular positive-strand RNA viruses, including rubella virus [71] and coronaviruses [72,73]. Indeed, C-to-U hypermutation has been observed and well-documented in both positive-strand RNA viruses [67,71,74], supporting our hypothesis. 

We are the first group to report the prevalence of HPgV-1 in HSCT pediatric patients, as well as the virus whole genomes and their intra-host diversity. The pathogenicity of HPgV-1 remains controversial, but a positive association of HPgV-1 viremia with the increased risk of lymphoma in immunocompromised individuals has been observed [75,76]. Although we could not establish a link between clinical symptoms and the detection of HPgV-1, further research should be undertaken to investigate the clinical impact of HPgV-1 on HSCT patients. Till then, the results of this study suggest that HSCT patients and their blood and stem cell donors should be routinely screened for HPgV-1 to reduce the risk of the virus transmission.

## Figures and Tables

**Figure 1 viruses-14-00796-f001:**
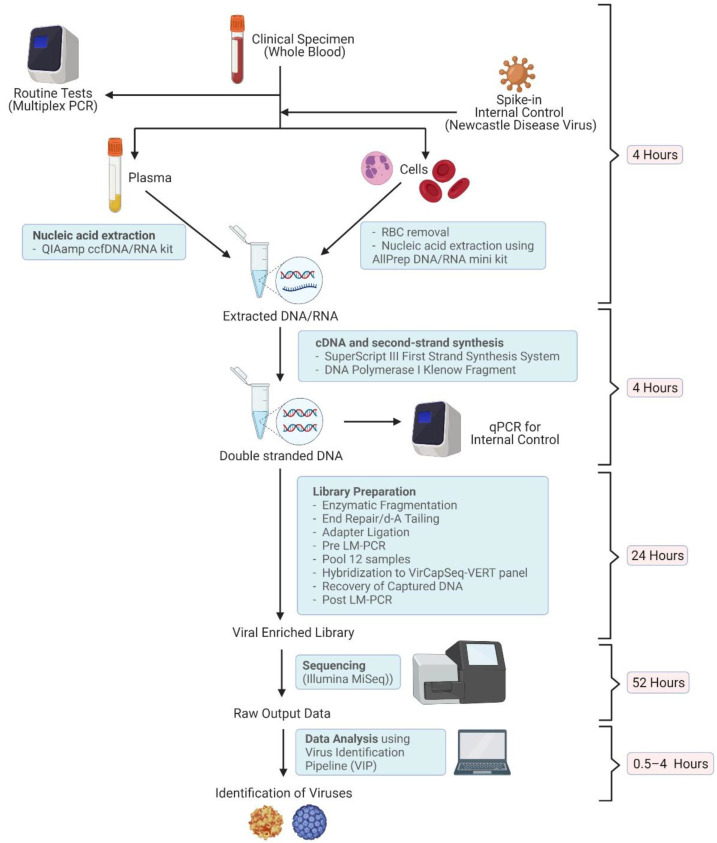
A schematic diagram of the virus detection procedure used in this study. Equipment and kits used, and the time consumed at each step are shown (Created with BioRender.com, accessed on 5 April 2022).

**Figure 2 viruses-14-00796-f002:**
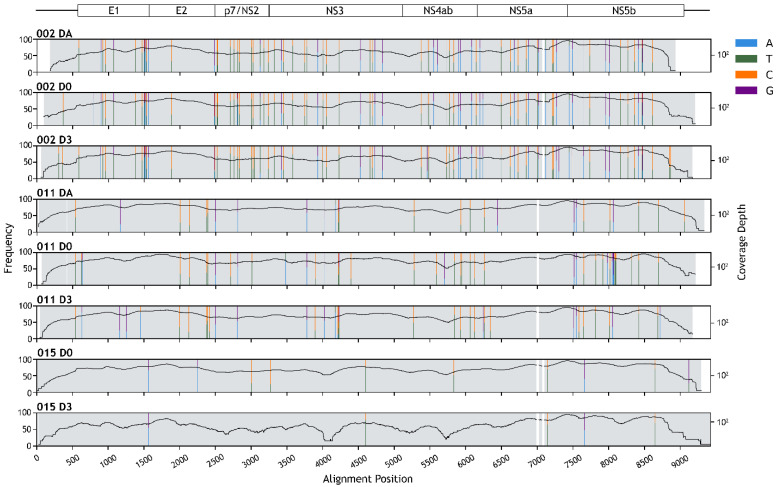
Human pegivirus-1 (HPgV-1) genome assembly. Schematic diagram of HPgV-1 genome organization is depicted at the top. Horizontal lines show untranslated regions, and rectangles show protein-coding regions. Positions of polymorphic sites are indicated by colored vertical stripes (see key for nucleotide variations). The height of the strips is proportional to the nucleotide frequency (left axis, linear scale). Black curves show the site-wise read depth of the genome assemblies (right axis, base-10 log scale). White regions indicate gaps in the sequence alignment.

**Figure 3 viruses-14-00796-f003:**
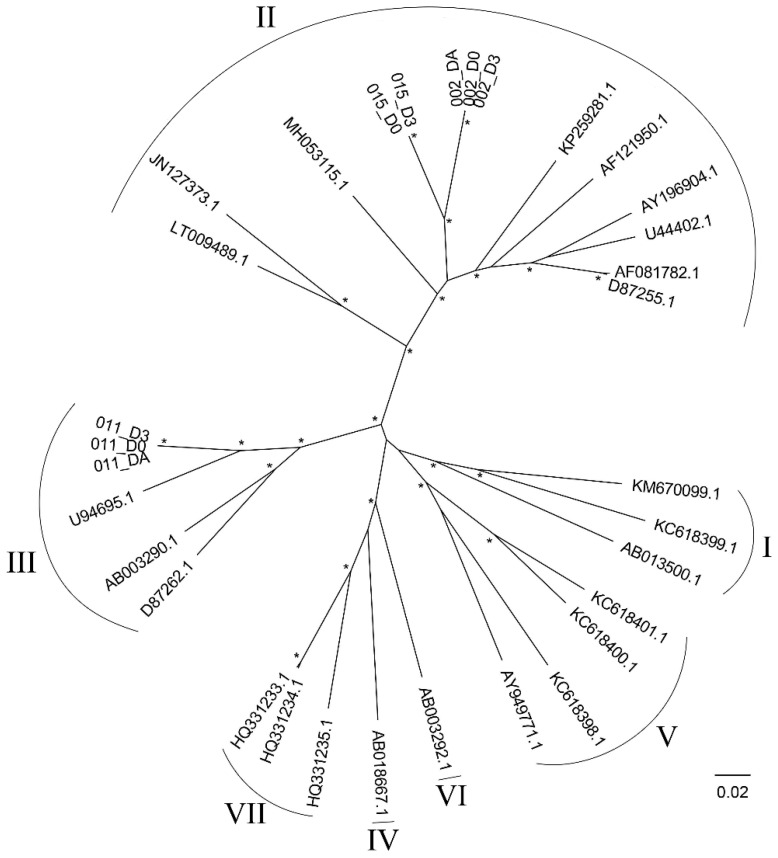
Maximum likelihood phylogenetic tree of Human pegivirus-1 (HPgV-1), genotypes 1 to 7. The phylogenetic tree was reconstructed by using IQ-TREE 1.6.12 [27] with the best-fit nucleotide substitution model (GTR + F + I + γ4) under the Bayesian information criterion. SH-aLRT and ultrafast bootstrap branch support was computed based on 10,000 bootstrapped trees. Genotypes of the viruses are indicated by Roman numerals. Asterisks (*) signify high (≥90%) bootstrap supported nodes. The scale bar is in the units of nucleotide substitutions per site.

**Table 1 viruses-14-00796-t001:** Summary of the associated clinical data of the samples sequenced in this study.

Sample Code	Gender	Age	Peak Body Temp (°C)	Fever Onset (Days)	Underlying Disease	Type of HSCT	ES	VOD	Prolonged Fever	GVHD	Conditioning Regimens	Long-Term (>2 Years) Follow-Up Data
001-D0	M	3Y6M	38.9	6	β thalassemia/ HbE	MR	No	No	No	No	BUSX+CTX	No long-term effects
001-D3			37.4	9								
002-DA	M	12Y1M	-	-	Acute myeloid leukemia	HAP	Yes	No	Yes (day 1–16)	No	FLU+THI+ BUSF	No long-term effects
002-D0			39.7	1								
002-D3			37.6	4								
003-D0	F	3Y7M	38.2	2	Neuroblastoma	HAP	Yes	No	No	Yes	BUSF+MEL+ATG	No long-term effects
003-D3			37.8	5								
004-D0	M	3Y2M	38.5	4	Langerhans cell Histiocytosis	HAP	No	No	Yes (day 1–19)	Yes	FLU+BUSF+ ATG	No long-term effects
004-D3			37	7								
005-D3	M	6Y2M	37.4	11	Anaplastic large cell lymphoma	AUT	No	No	No	No	BCNU+ETO+CTX	No long-term effects
006-D0	F	3Y8M	40	6	Yolk sac tumor	AUT	No	No	No	No	ETO+CARB+CTX	No long-term effects
006-D3			37.1	9								
007-D0	M	1Y11M	38.2	2	Acute myeloid leukemia	HAP	No	No	No	Yes	FLU+THI+ BUSF	No long-term effects
007-D3			37.4	5								
008-D0	M	3Y9M	39.7	10	Chronic granulomatous disease	HAP	No	No	Yes (day 8–18)	Yes	FLU+BUSF+ ATG	Chronic GVHD
008-D3			38.4	13								
009-D0	F	3Y9M	40	3	Neuroblastoma	HAP	No	No	No	Yes	BUSF+MEL+ATG	No long-term effects
009-D3			36.6	6								
011-DA	M	2Y4M	-	-	Wiskott Aldrich Syndrome	MR	No	No	No	No	BUSF+CTX	No long-term effects
011-D0			38.1	3								
011-D3			38.5	7								
012-D0	F	1Y4M	38.9	2	Acute lymphocytic leukemia	HAP	Yes	No	No	No	FLU+THI+ BUSF	No long-term effects
012-D3			38.0	6								
013-D0	F	3Y5M	38.2	3	β thalassemia/ HbE	HAP	Yes	No	No	Yes	FLU+BUSF+ ATG	Death from GVHD and bleeding
013-D3			38.3	7								
014-DA	F	3Y3M	-	-	Undifferentiated round cell tumor	AUT	-	-	No	-	BUSX+MEL	Death from RDS on 39 days post-HSCT
015-DA	M	7Y6M	-	-	Neuroblastoma	MR	No	No	No	No	BUSF+MEL	Neuroblastoma relapse on 164 days post-HSCT
015-D0			38.1	6								
015-D3			37.6	9								

M = male, F = female, HbE = hemoglobin E, MR = match related, HAP = haploidentical, AUT = autologous, ES = engraftment syndrome, VOD = veno-occlusive disease, GVHD = graft versus host disease, CTX = cyclophosphamide, ATG = anti-thymocyte globulin, BUSX = bulsulflex, BUSF = busulfan, MEL = melphalan, FLU = fludarabine, THI = thiotepa, CARB = carboplatin, ETO = etoposide, RDS = respiratory distress syndrome.

**Table 2 viruses-14-00796-t002:** Summary of the viral sequences detected in the target enrichment next-generation sequencing (NGS) data using Virus Identification Pipeline, VIP [16].

TE-NGS											Routine Test	Confirmation
Sample Code	Total Number of Reads (Million)	Viral Reads %	Total Number of Viral Reads	NDV-Internal Control			Identified Viruses					
				% Genome Coverage	Reads	% ^a^	Virus Identified	% Genome Coverage	Reads	% ^a^	Multiplexed qRT-PCR ^b^	Targeted qRT-PCR ^b^
001-D0	1.83	21.4	391,620	100	139,882	35.7	-	-	-	-	Neg	NA
001-D3	3.76	48.4	1,819,840	100	1,737,559	95.5	-	-	-	-	Neg	NA
002-DA	2.66	91.8	2,441,880	100	2,384,116	97.6	HPgV-1	94.2	33,916	1.4	Neg	HPgV-1 (Ct 28.58)
002-D0	2.11	28.5	601,350	100	362,876	60.3	HPgV-1	98.4	31,480	0.6	Neg	HPgV-1 (Ct 29.10)
002-D3	4.02	59.6	2,395,920	100	2,302,948	96.1	HPgV-1	96.5	23,319	0.9	Neg	HPgV-1 (Ct 29.54)
003-D0	2.46	32.0	787,200	100	741,509	94.2	-	-	-	-	Neg	NA
003-D3	1.59	10.6	168,540	100	5415	3.2	-	-	-	-	Neg	NA
004-D0	3.91	58.8	2,299,080	100	2,232,001	97.1	-	-	-	-	CMV (Ct 34.07)	NA
004-D3	0.73	5.5	40,150	100	21,688	54.0	CMV	67.2	2221	5.5	CMV (Ct 31.18)	NA
005-D0	1.90	16.2	307,800	100	268,477	87.2	-	-	-	-	Neg	NA
006-D0	1.71	2.3	39,330	99.5	3174	8.0	-	-	-	-	Neg	NA
006-D3	2.63	42.8	1,125,640	100	1,057,996	94.0	-	-	-	-	Neg	NA
007-D0	1.97	13.7	269,890	100	69,031	25.6	-	-	-	-	Neg	NA
007-D3	1.05	82.1	862,050	100	841,928	97.7	-	-	-	-	Neg	NA
008-D0	2.06	6.8	140,080	100	76,091	54.3	-	-	-	-	CMV (Ct 35.81)	NA
008-D3	2.19	2.8	61,320	99.6	7035	11.5	-	-	-	-	CMV (Ct 36.89)	NA
009-D0	1.97	3.2	63,040	100	12,533	19.9	-	-	-	-	Neg	NA
009-D3	1.27	2.5	31,750	99.5	2916	9.2	-	-	-	-	Neg	NA
011-DA	5.36	69.8	3,741,280	100	1,740,886	46.5	HPgV-1	100	69,446	1.8	Neg	HPgV-1 (Ct 25.17)
011-D0	1.86	36.8	684,480	100	603,461	88.2	HPgV-1	98.9	44,268	6.4	Neg	HPgV-1 (Ct 30.22)
011-D3	2.20	38.9	855,800	100	758,192	88.6	HPgV-1	99.0	62,142	7.2	Neg	HPgV-1 (Ct 30.65)
012-D0	4.61	72.1	3,323,810	100	3,165,402	95.2	-	-	-	-	Neg	NA
013-D0	1.49	12.7	189,230	100	138,532	73.2	-	-	-	-	Neg	NA
013-D3	1.70	19.4	329,800	100	102,209	31.0	-	-	-	-	Neg	NA
014-DA	TE-NGS was not performed										Neg	Neg for HPgV-1
015-DA	1.04	15.5	161,200	100	85,325	52.9	-	-	-	-	Neg	Neg for HPgV-1
015-D0	3.40	59.4	2,019,600	100	830,550	41.1	HPgV-1	97.9	58,016	2.8	Neg	HPgV-1 (Ct 26.94)
015-D3	0.19	43.9	83,410	100	30,992	37.1	HPgV-1	98.6	3935	4.7	Neg	HPgV-1 (Ct 26.28)

^a^ Percentage of detected viral read in total number of total viral reads, including those of the positive control NDV. ^b^ The Ct values from the two columns are not comparable due to the fact that the Ct values reported for the routine tests (left column) were estimated from probe-based multiplex qRT-PCR assays, while those from the targeted PCR assays (right column) were estimated using SYBR-Green-based assays. Ct values from the same column, however, could be compared. NA = not applicable, Neg = negative.

**Table 3 viruses-14-00796-t003:** Human pegivirus-1 (HPgV-1) genome assembly statistics.

Sample Name	Genome Size	Number of Reads	Average Assembly Depth	Accession Number
002-DA	8714	66,577	1436	MZ099565
002-D0	9082	54,861	1273	MZ099566
002-D3	9080	40,980	914	MZ099567
011-DA	9273	118,572	2764	MZ099568
011-D0	9112	77,867	1722	MZ099569
011-D3	9096	106,570	2485	MZ099570
015-D0	9232	98,765	2268	MZ099571
015-D3	9336	6691	154	MZ099572

## Data Availability

NGS sequencing data generated by this study are available from the Sequence Read Archive (SRA) repository under the project accession number PRJNA705169. All consensus HPgV-1 genome sequences generated by this study are available in GenBank with the following accession numbers: MZ099565 to MZ099572.

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
