# Peer review of "Target Enrichment Metagenomics Reveals Human Pegivirus-1 in Pediatric Hematopoietic Stem Cell Transplantation Recipients"

_viruses, 2022, doi:10.3390/v14040796_

Round 1
Reviewer 1 Report
the paper has been substantially reworked, my principal concern about the lack of control population has been adressed by studying healthy adolescent's samples.
the authors have responded satisfactorily to the remarks of the 3 reviewers and thus I judge that the paper is now suited for publication.
The only remaining problem is the table 1 which is difficult to read because of the high number of columns and truncated words, especially the chemotherapy drugs. I suggest to suppress two columns, one including the white cell count which is of poor interest and the one regarding VOD, since there was no VOD case, thus the information can be given in the text. Regarding the drugs, I suggest abbreviations such as:BU, Flu, Thio, Eto, Carbo ... and to give the full names under the table
Author Response
Dear Reviewer #1, The authors would like to thank you for the valuable comments that helped to improve the manuscript. All the points that you have made have been addressed below:
Comment 1: the paper has been substantially reworked, my principal concern about the lack of control population has been addressed by studying healthy adolescent's samples. The authors have responded satisfactorily to the remarks of the 3 reviewers and thus I judge that the paper is now suited for publication.
Response: Thank you very much.
Comment 2: The only remaining problem is the table 1 which is difficult to read because of the high number of columns and truncated words, especially the chemotherapy drugs. I suggest suppressing two columns, one including the white cell count which is of poor interest and the one regarding VOD, since there was no VOD case, thus the information can be given in the text. Regarding the drugs, I suggest abbreviations such as:BU, Flu, Thio, Eto, Carbo ... and to give the full names under the table.
Response: Thank you for pointing this out. We have edited the table accordingly. However, we have left the VOD column, because we believe that it will be easier for the readers to get a comprehensive idea on the short-term clinical complications if we leave all four of them side by side.
Reviewer 2 Report
Ludowyke et al. showed the necessity to screen HSCT patients and blood and stem cell donors to reduce the potential risk of HPgV-1 transmission. In current study, The SeqCap EZ VirCapSeq-VERT Panel was used to search for viruses present in the blood of pediatric patients undergoing HSCT.  Human pegivirus was identified in three patients, but no significant association between the presence of the virus and the pathology was demonstrated. In addition, detailed genomic analysis of the identified pegiviruses was also conducted. Current paper is potentially interesting, but several modifications are needed for publication. My concerns are listed below.
Comments
- Figure 1 is difficult to read due to low resolution and small text. Please make the text larger and make the figure easier to understand.
- Although the present method can only search for known viruses, there is a possibility that there are unknown viruses involved in the pathogenesis. Various methods of searching for unknown viruses have been reported, and such papers should be cited to discuss the importance of searching for unknown viruses.
- Pegiviruses are unique in that they do not have a core protein in the viral genome. The authors should consider whether there are any sequences in Contig obtained from this genome analysis that could serve as core proteins of Pegivirus.
- The virus discovery method using the SeqCap EZ VirCapSeq-VERT Panel is considered a revolutionary experimental system. Are there any reports on studies using this method so far? If there is one, the authors should provide a discussion of that report.
Author Response
Dear Reviewer #2, The authors would like to thank you for the valuable comments that helped to improve the manuscript. All the points that you have made have been addressed below:
Comment 1: Ludowyke et al. showed the necessity to screen HSCT patients and blood and stem cell donors to reduce the potential risk of HPgV-1 transmission. In current study, The SeqCap EZ VirCapSeq-VERT Panel was used to search for viruses present in the blood of pediatric patients undergoing HSCT. Human pegivirus was identified in three patients, but no significant association between the presence of the virus and the pathology was demonstrated. In addition, detailed genomic analysis of the identified pegiviruses was also conducted. Current paper is potentially interesting, but several modifications are needed for publication. My concerns are listed below. Figure 1 is difficult to read due to low resolution and small text. Please make the text larger and make the figure easier to understand.
Response: Thank you very much for your comment. As you have suggested, we have done the necessary changes to Figure 1.
Comment 2: Although the present method can only search for known viruses, there is a possibility that there are unknown viruses involved in the pathogenesis. Various methods of searching for unknown viruses have been reported, and such papers should be cited to discuss the importance of searching for unknown viruses.
Response: Thank you for raising this valuable point. In fact, the TE-NGS platform used in this study can be used to detect unknown viruses. Although the VirCapSeq-VERT system is not specifically designed for viral discovery, the inventors of this probe panel have shown its potential to detect novel viruses with an overall nucleotide divergence in the range of 40% (Briese et. al, 2015). This is enabled by hybridization of probes to short conserved sequence motifs within larger genome fragments.
This point has been clarified in the manuscript and quoted along with the response to your comment #4.
Comment 3: Pegiviruses are unique in that they do not have a core protein in the viral genome. The authors should consider whether there are any sequences in Contig obtained from this genome analysis that could serve as core proteins of Pegivirus.
Response: Thank you for the interesting question. We’ve obtained complete genome sequences of HPgV-1’s identified in all the 3 patients by de novo assembly, thus the assembly process was not influenced by the fact that the reference genomes in databases don’t contain capsid protein-coding regions. Our de novo assembled sequences were 94.2–100% similar to the existing HPgV-1 genome sequences present in databases, and we didn’t see any additional protein-coding regions (corresponding to capsid/core proteins) in any of the identified HPgV-1’s.
Comment 4: The virus discovery method using the SeqCap EZ VirCapSeq-VERT Panel is considered a revolutionary experimental system. Are there any reports on studies using this method so far? If there is one, the authors should provide a discussion of that report.
Response: Thank you for the question. Yes, the developers of this panel have extensively studied the performance of VirCapSeq-VERT Panel (Briese et. al, 2015). Later on, a few more studies have used this panel to explore the virome of different sample types. However, the TE-NGS platform of this study was developed and optimized in-house, and the methodology used from the sample to answer is not identical to the original paper or subsequent papers. For example, the RBC removal step, the extraction kits used, the internal control, and the bioinformatic analysis approach are unique to the TE-NGS platform described in this study.
The following sentence was added to the second paragraph of the Discussion, and it includes the answer to the 2nd point raised by reviewer #2:
The VirCapSeq-VERT is a probe panel comprised of ~2 million probes that cover all the viruses known to infect vertebrates. Its potential to enrich viral sequences of known viruses up to 10,000 folds when compared to conventional high throughput sequencing, and to detect novel viruses with an overall nucleotide divergence in the range of 40% have been demonstrated [15]. This makes this probe panel suitable for detecting divergent and potentially unknown viruses in clinical samples, which is essential when it comes to analyzing clinical samples from patients with unexplained illnesses, and for outbreak investigation of both known and unknown infectious etiologies. Subsequent other studies have also utilized this panel to explore the virome of a range of clinical and environmental samples and have obtained compelling results [46-50].
Reviewer 3 Report
While HPgV-1 infection is not persuasively corrected to known human disease, this study reports the prevalence of HPgV-1 in HSCT pediatric patients (3 out 14 detectable), suggesting the importance of the screening to reduce the risk of transmission. The conclusion was based on the results from targeted enrichment NGS. As the existence of viral RNA in the plasma is usually corrected to active infection, was the virus in the patient samples also detectable by routine qRT-PCR in the clinical setting? Authors might clarify this point.
The experiments were well designed, data analyses were reliable and the discussion section was well written. Reviewer recommends its publication in Viruses.
Author Response
Dear Reviewer #3, The authors would like to thank you for the valuable comments that helped to improve the manuscript. All the points that you have made have been addressed below:
Comment 1: The experiments were well designed, data analyses were reliable, and the discussion section was well written. Reviewer recommends its publication in Viruses.
Response: Thank you very much.
Comment 2: While HPgV-1 infection is not persuasively corrected to known human disease, this study reports the prevalence of HPgV-1 in HSCT pediatric patients (3 out 14 detectable), suggesting the importance of the screening to reduce the risk of transmission. The conclusion was based on the results from targeted enrichment NGS. As the existence of viral RNA in the plasma is usually corrected to active infection, was the virus in the patient samples also detectable by routine qRT-PCR in the clinical setting? Authors might clarify this point.
Response: Thank you for the comment. HPgV-1 is not covered by the routine qRT-PCR in the clinical settings of this patient group, and that has been mentioned in the text of the introduction, results, and discussion sections.
By the confirmatory qRT-PCR (which we did after detecting HPgV-1 by TE-NGS), we were able to detect HPgV-1 in the plasma of all the samples that were also detected positive by TE-NGS. Therefore, as per your suggestion, we highlighted the fact that it could be suggestive of an active infection:
In Results (page 7): The detections of HPgV-1 by bioinformatic analyses were confirmed by subjecting plasma samples to qRT-PCR using HPgV-1 specific primers [29], yielding Ct values ranging between 25.17 and 30.65 (Table 2). The fact that HPgV-1 could be detected in the plasma is suggestive of an active infection.
This manuscript is a resubmission of an earlier submission. The following is a list of the peer review reports and author responses from that submission.
Round 1
Reviewer 1 Report
In this work, the authors have explored 15 pediatric recipients of HSCT with febrile neutropenia. They found frequent expression of HPgV, namely 3 of the 15 patients had circulating HPgV RNA.
The authors have analysed the variability and the subtypes of the viruses. Tje quality of the work is highlighted in the clear figures and tables, the potential of the methodology of viral genome detection is evidenced.
However, there are frailties in the demonstration; one major problem relies in the absence of controle group, namly HSCT recipients without febrile neutropenia. Thus we are not able to establish wether there is a correlation between fever and HPgV RNA isolation. The normal results of liver tests is somehow reassuring but we have no long term follow-up, in order to establish wether, like for CMV, there might be a role of HPgV infection on post-transplant relapse risk.
From reading the manuscript, I could not clearly understand if other viruses have been identified in the patients through this screening. HHV6 reactivation has for instance been described to occur in two thirds of patients (P.J.Anne de Pagter, BBMT July 2008, Pages 831-839).
The clinical impact of the findings should be discussed in more details in the discussion. The duration of the viral load positivity should be investigated. The points mostly discussed in details relate to the virus variabilities, to my mind, there is no clinical relevance of comparing two viral samples distant from 3 days.
Regarding the references, some of them could be ommited without impairing the manuscript.
Reviewer 2 Report
The text by Ludowyke N et al. it is interesting because it describes 3 cases of HPgV in the setting of stem cell transplantation in pediatric patients. The introduction is clear and concise as well as the materials and patients part is well described. In the results section the authors should report
1) the co-presence of other viruses that could significantly impact the results of laboratory tests or declare negativity at the time DA DO and D3 of the main viruses such as CMV, EBV, AdenoV etc, etc.
2) the authors should also report the outcome of patients with HPgV and, in positive cases, also report, if possible, the negativization of the viral load and the timing. The TE-NGS section is well described and the conclusions are consistent with the results obtained.
Reviewer 3 Report
This manuscript describes the detection and intra-host diversity of human pegivirus-1 (HPgV-1) in pediatric HSCT recipients. The paper is well written and the analyses comprehensive, but I believe that few points should be revised to make the text suitable for publication.
Please, see below suggestions that may help clarify some points.
- Introduction: Currently, two types of pegivirus infecting humans are recognized. This manuscript is supposed to be about HPgV-1 (species Pegivirus C), but this information isn’t in the text.
- Tables: Please, consider reformating the tables so column ID and words in cells are not separated because of the reduced width of each column.
- Page 6, item 3.2: What was the average time of onset of febrile condition? Is there any difference in FN onset between HPgV-1 positive and negative individuals?
- Page 9, item 3.5: I suggest the authors remove this item. It is already well defined that HPgV-1 infection is not associated with liver injury, so this table and text don’t aggregate any relevant information to the manuscript.
- Discussion: The finding of co-circulation of genotypes 2 and 3 is not exactly something surprising since genotype 2 is widely spread, maybe the most ubiquitous pegivirus genotype.
- Discussion: I suggest considering the importance of monitoring the presence of a non-pathogenic virus in immunosuppressed individuals, which can be consulted in Vu et al 2019.